# Post-Transplant Cyclophosphamide-Based Prophylaxis and Its Impact on Infectious Complications and Immune Reconstitution According to Donor Type

**DOI:** 10.3390/cancers17071109

**Published:** 2025-03-26

**Authors:** Beatriz Merchán-Muñoz, María Suárez-Lledó, Luis Gerardo Rodríguez-Lobato, Tommaso Francesco Aiello, Antonio Gallardo-Pizarro, Paola Charry, Joan Cid, Miquel Lozano, Alexandra Pedraza, Alexandra Martínez-Roca, Ares Guardia, Laia Guardia, Cristina Moreno, Enric Carreras, Laura Rosiñol, Carolina García-Vidal, Francesc Fernández-Avilés, Carmen Martínez, Montserrat Rovira, María Queralt Salas

**Affiliations:** 1Hematopoietic Transplantation Unit, Hematology Department, Institute of Cancer and Blood Diseases (ICAMS), Hospital Clínic Barcelona, 08036 Barcelona, Spain; msuarezl@clinic.cat (M.S.-L.); lgrodriguez@clinic.cat (L.G.R.-L.); apmartinez@clinic.cat (A.M.-R.); aguardiat@clinic.cat (A.G.); lguardia@clinic.cat (L.G.); cmoreno1@clinic.cat (C.M.); lrosinol@clinic.cat (L.R.); ffernand@clinic.cat (F.F.-A.); cmarti@clinic.cat (C.M.); mrovira@clinic.cat (M.R.); 2Institut d’Investigacions Biomèdiques August Pi i Sunyer (IDIBAPS), 08036 Barcelona, Spain; 3Department of Infectious Disease, Hospital Clínic Barcelona, 08036 Barcelona, Spain; tfaiello@recerca.clinic.cat (T.F.A.); agallardo@clinic.cat (A.G.-P.); cgarciav@clinic.cat (C.G.-V.); 4Apheresis and Cellular Therapy Unit, Hemotherapy and Hemostasis Department, Institute of Cancer and Blood Diseases (ICAMS), Hospital Clínic de Barcelona, 08036 Barcelona, Spain; charry@clinic.cat (P.C.); jcid@clinic.cat (J.C.); mlozano@clinic.cat (M.L.); acpedraza@clinic.cat (A.P.); 5Josep Carreras Institute Against Leukemia, Campus Clinic, 08916 Badalona, Spain; enric.carreras@fcarreras.es; 6Faculty of Medicine and Health Sciences, University of Barcelona, 08036 Barcelona, Spain

**Keywords:** PTCY-based prophylaxis, allo-HCT, immune reconstitution, infection density, infectious complications

## Abstract

This study evaluated the impact of donor type on infectious complications and immune reconstitution in adults undergoing allogeneic hematopoietic cell transplantation with post-transplant cyclophosphamide-based prophylaxis. A total of 253 patients were included and stratified by donor source: HLA-matched, mismatched unrelated, or haploidentical. Most infections occurred within the first 100 days post-transplant, predominantly bacterial bloodstream infections and viral reactivations. Notably, infection rates and clinical outcomes were comparable across donor groups. By six months post-transplant, most patients showed signs of immune recovery. Importantly, long-term outcomes were similar regardless of the donor source. These findings suggest that post-transplant cyclophosphamide-based prophylaxis is generally safe and effective across different donor types, and provide valuable insights for optimizing supportive care and infection monitoring in hematopoietic stem cell transplant recipients.

## 1. Introduction

The use of post-transplant cyclophosphamide (PTCY) combined with additional immunosuppressive agents for graft-versus-host disease (GVHD) prevention has gained widespread acceptance in allogeneic hematopoietic cell transplantation (allo-HCT). PTCY’s action resides in inducing apoptosis of rapidly proliferating alloreactive T-cells while sparing regulatory T-cells, resulting in the rapid suppression of quantitative immune reconstitution [1]. This approach has demonstrated efficacy in promoting stem cell engraftment and preventing GVHD, initially in haploidentical donor transplants (haplo-HCT) and later in transplants regardless of donor type [2,3,4,5].

Despite these benefits, the increasing adoption of PTCY-based prophylaxis has been linked to delayed engraftment, impaired immune reconstitution, and a higher incidence of infectious complications, which remain a leading cause of mortality in allo-HCT patients [6,7,8]. While prior studies have examined the relationship between PTCY and infection risk, differences in infectious complication rates based on donor type have not been thoroughly explored.

At our institution, PTCY-based prophylaxis was introduced for haplo-HCT in 2013, demonstrating effective GVHD prevention. The experience gained from its use demonstrated its efficacy for GVHD prevention, leading to its adoption as our institutional prophylaxis for all allo-HCT performed irrespective of donor type [9,10]. Given the potentially life-threatening nature of infectious complications, this study investigates the incidence of these events and immune reconstitution dynamics in allo-HCT recipients from a single center, with a focus on donor type.

## 2. Materials and Method

### 2.1. Patient Selection

In this study, we analyzed 253 consecutive adult patients who underwent peripheral blood (PB) allo-HCT with PTCY-based prophylaxis at our institution between January 2013 and December 2021. All patients were diagnosed with hematologic malignancies and received peripheral blood stem cell (PBSC) grafts. The data were retrospectively collected and last updated in June 2023. The study adhered to the ethical standards outlined in the Declaration of Helsinki and was approved by the Ethics Committee of the Hospital Clinic de Barcelona.

### 2.2. Transplant Information and GVHD Prophylaxis

Eligibility criteria for allo-HCT and donor selection algorithms are detailed in the Appendix A. Reduced-intensity conditioning (RIC) regimens were generally reserved for patients older than 55 years, those with prior HCT, or those with significant comorbidities. Myeloablative conditioning (MAC) regimens typically included fludarabine (30 mg/m^2^/day for 4 days) combined with either high-dose busulfan (3.2 mg/kg/day for 4 days) or 12 Gy of total body irradiation (TBI). RIC regimens included reduced doses of busulfan (3.2 mg/kg/day for 3 days) or 8 Gy of TBI, combined with standard fludarabine dosing. Patients undergoing haplo-HCT received 2 Gy of TBI if it was not already part of the conditioning regimen.

All patients received PTCY at 50 mg/kg on days +3 and +4 post-transplant, combined with tacrolimus (TK) initiated on day +5 at a dose of 0.03 mg/kg/24 h. Tacrolimus levels were maintained between 5 and 15 ng/mL until day +90 and tapered by day +180 in the absence of GVHD or relapse. Haplo-HCT recipients also received mycophenolate mofetil from day +5 to day +35. No patient received antithymocyte globulin. T-cell-replete PBSC grafts, without prior manipulation, were infused on day 0. Granulocyte colony-stimulating factor (G-CSF) was not administered throughout the study.

Engraftment after allo-HCT was defined as the presence of an absolute neutrophil count greater than ≥0.5 × 10^9^/L on the first of three consecutive days. Platelet recovery was defined as a sustained platelet count > 20 × 10^9^/L (1st of 3 days) without platelet transfusion for 7 days.

### 2.3. Infectious Prophylaxis, Monitoring, Treatment, and Main Definitions

The institutional antimicrobial prophylaxis regimen remained unchanged throughout the study and included levofloxacin 500 mg daily from day 1 until neutrophil engraftment, fluconazole 400 mg daily from day 1 to day 60, acyclovir 800 mg twice daily from day 1 up to one year post-allo-HCT, and either trimethoprim-sulfamethoxazole 160/800 mg three times per week or inhaled pentamidine 300 mg monthly until peripheral blood CD4+ cell counts exceeded 200 cells/mL [11]. No patient received letermovir for cytomegalovirus (CMV) prophylaxis.

Infection monitoring followed institutional guidelines. Weekly serum aspergillus galactomannan antigen (AGA) and serum CMV quantitative PCR monitoring were performed routinely through day +100 after allo-HCT in all patients. Infectious complications were managed homogeneously during the study period following standard practices [12,13,14,15].

### 2.4. Definitions and Supportive Care

Bacterial bloodstream infection (BSI) was defined as the isolation of a bacterial pathogen from at least one set of blood cultures (one aerobic and one anaerobic) collected from a patient post-allo-HCT. BSIs due to common skin colonizers, including coagulase-negative staphylococci and Corynebacterium, were considered clinically significant only if accompanied by a suggestive clinical picture and either ≥2 positive blood cultures from distinct sites or one positive peripheral vein blood culture alongside a positive catheter culture. Contaminants were excluded from the infected patient cohort. Polymicrobial infection was defined as a BSI involving two or more microbial species.

CMV, human herpesvirus-6 (HHV6), and BK polyomavirus infections were classified as viremia or disease using BMT CTN infection reporting criteria [16,17]. CMV reactivation was defined as the presence of CMV DNAemia in peripheral blood, with levels of ≥500 copies detected by quantitative PCR or ≥650 IU/mL by real-time PCR [18]. Epstein–Barr virus (EBV) reactivation was characterized by EBV-DNA levels of ≥1000 copies/mL. HHV6 diagnosis was established using quantitative PCR on peripheral blood or tissue biopsy samples [19].

### 2.5. Statistical Methods

Demographics, transplant characteristics, and immune reconstitution metrics were summarized using descriptive statistics. Categorical variables were compared using χ^2^ or Fisher’s exact tests, while continuous variables were assessed with the Wilcoxon test. Data were presented as counts and percentages for categorical variables and medians with ranges for continuous variables.

Event times were calculated from the date of stem cell infusion (day 0) to the date of the event or last follow-up. The cumulative incidence (Cum.Inc) of infectious complications was estimated using the cumulative incidence method, considering death as a competing risk, and was reported at days +30, +100, +180, and one year post-transplant. Infection density was calculated by dividing the total number of infections by the observed post-transplant period. Immune reconstitution was monitored by measuring levels of immunoglobulin G (IgG), CD4+ T-cells, and CD8+ T-cells in relapse-free surviving patients. Overall Survival (OS) was calculated using the Kaplan–Meier method. Non-relapse mortality (NRM) was estimated using the cumulative incidence method with Gray’s test, considering relapse as a competing event. All *p* values were two-sided, with *p* < 0.05 considered statistically significant. Statistical analysis was performed using EZR [20].

## 3. Results

### 3.1. Patients and Allo-HCT Baseline Information

The main characteristics of the study cohort, stratified by donor type, are summarized in Table 1. The median age of the patients was 53 years (range, 18–70), with 44 (17.6%) aged over 65 years. Acute myeloid leukemia (AML, *n* = 90, 36%) and myelodysplastic syndrome (MDS, *n* = 43, 18%) were the most prevalent underlying diagnoses. A total of 103 (41%) patients underwent myeloablative conditioning (MAC) allo-HCT, and all patients received peripheral blood stem cell (PBSC) grafts

As presented in Table 1, 120 patients (47.4%) received grafts from HLA-matched donors, 84 (33.2%) from 9/10 HLA-mismatched unrelated donors (MMUD), and 49 (19.4%) from haploidentical donors. Among the 10/10 HLA-matched donor–recipient pairs, 28 patients (23.3%) received grafts from 10/10 HLA-matched sibling donor, while 92 patients (72.7%) underwent transplantation with 10/10 HLA-matched unrelated donor (MUD). Baseline characteristics were similar across the three groups, except for a higher frequency of patients with an HCT-CI > 3 in the HLA-matched donor group (31.7% vs. 17.9% vs. 16.3%, *p* = 0.028).

### 3.2. Main Post-Transplant Information and Outcomes

As shown in Table 2, engraftment occurred in 241 (95.3%) patients. The median time to neutrophil and platelet engraftment was 19 days (IQR: 11–103) and 18 days (IQR: 6–141), respectively, with no significant differences based on donor type. Primary graft failure (GF) occurred in 12 (4.7%) patients: 5 (4.1%) in the HLA-matched group, 6 (7.1%) in the MMUD group, and 1 (0.02%) in the haploidentical group. Among these patients, 8 (66.7%) underwent a second allograft, achieving engraftment, though the associated mortality rate was 62.5%.

The cumulative incidence (Cum.Inc) of grade II–IV and grade III–IV aGVHD at day +100 was 19.2% (95% CI, 12.7–26.7) and 6.7% (95% CI, 3.1–12.1) in the HLA-matched group, 31.0% (95% CI, 21.4–41.0) and 8.3% (95% CI, 3.6–15.5) in the MMUD group, and 22.4% (95% CI, 11.9–35.0) and 4.1% (95% CI, 0.7–12.5) in the haploidentical group (*p* = 0.055 and *p* = 0.848, respectively). The 2-year Cum.Inc of moderate/severe cGVHD was 3.6% (95% CI, 1.2–8.4), 13.6% (95% CI, 6.9–22.6), and 9.6% (95% CI, 3.0–21.0) in the HLA-matched, MMUD, and haploidentical groups, respectively (*p* = 0.062).

With a median follow-up of 29.4 months (IQR: 5.52–68), 71 (28.1%) patients relapsed, and 96 (37.9%) died. The main causes of death were relapse and infections, with no significant differences across donor types. As illustrated in Figure 1, the estimated 2-year OS was 68.1% (95% CI, 58.5–75.9) for HLA-matched donors, 61.6% (95% CI, 50.3–71.1) for MMUD, and 66.8% (95% CI, 51.6–78.2) for haploidentical donors (*p* = 0.759). The estimated 2-year NRM was 11.8% (95% CI, 6.8–18.4), 26.4% (95% CI, 17.4–36.3), and 22.4% (95% CI, 11.9–35.0), respectively (*p* = 0.0528), shown in Figure 2.

### 3.3. Incidence of Infectious Complications According to Donor Type

Table 2 highlights the incidence of bloodstream infections (BSI), which were predominantly diagnosed during the initial 30 days post-transplant. The day +30 Cum.Inc of BSI was 49.2% (95% CI, 39.9–57.8) in the HLA-matched group, 38.1% (95% CI, 27.7–48.4) in the MMUD group, and 34.7% (95% CI, 21.7–48.0) in the haploidentical group (*p* = 0.073).

CMV reactivation primarily occurred between days +30 and +100, with a median onset of 61 days (IQR: 5–3190) post-transplant. The day +100 Cum.Inc of CMV reactivation was 39.2% (95% CI, 30.4–47.8) in the HLA-matched group, 59.5% (95% CI, 48.1–69.2) in the MMUD group, and 55.1% (95% CI, 40.0–67.9) in the haploidentical group (*p* = 0.033). CMV disease was less frequent, with gastrointestinal involvement tending to be the most common manifestation. The day +100 Cum.Inc of CMV disease was 5.8% (95% CI, 2.6–11.0), 6.0% (95% CI, 2.2–12.4), and 6.1% (95% CI, 1.6–15.3) in the HLA-matched, MMUD, and haploidentical groups, respectively (*p* = 0.449).

HHV-6 reactivation or disease showed a trend toward higher incidence rates in patients receiving haploidentical donor grafts (*p* = 0.068), with a day +100 Cum.Inc of 7.5% (95% CI 3.7–13.1) in the HLA-matched group, 9.5% (95% CI 4.4–17.0) in the MMUD group, and 22.4% (95% CI 11.9–35.0) in the haploidentical group.

Grade 2–4 BK-virus-associated hemorrhagic cystitis demonstrated a trend toward higher incidence in patients undergoing MMUD allo-HCT (*p* = 0.056). The day +100 Cum.Inc was 7.6% (95% CI 3.7–13.2) for the HLA-matched group, 22.3% (95% CI 13.9–32) for the MMUD group, and 14.8% (95% CI 6.4–26.5) for the haploidentical group.

Respiratory tract infections were the second most prevalent viral complication, with their incidence steadily increasing during the first year post-allo-HCT. Rhinovirus was the most frequently detected pathogen, accounting for 32.4% of respiratory viral infections, followed by SARS-CoV2, which was responsible for 26.8%.

Fungal infections occurred in 12.7% of patients, predominantly within the first 100 days post-transplant. Pulmonary aspergillosis (62.8%) and oropharyngeal candidiasis (18.6%) were the most frequent presentations, with no significant differences among donor groups (*p* = 0.640).

All patients underwent specific treatment, but 17 (6.7%) adults died secondary to the diagnosis of these complications. More specifically, the day +30 mortality rate of patients with first BSI was 2.4%, that of patients with fungal infection was 3.16%, that of patients with respiratory tract infections was 0.79%, and that of patients with CMV disease was 0.39%.

### 3.4. Infectious Density

The analysis of infection density revealed a comparable incidence of infectious complications among the three study groups. As illustrated in Figure 3, infection density was notably higher during the initial 100 days following allo-HCT, with most patients experiencing one or two infections within this period. Beyond day +100, infection density progressively decreased throughout the post-transplant follow-up.

When data were stratified by donor type, infection density within the first 100 days post-transplant varied slightly. In the HLA-matched group, 21.7% of patients had no infections, 43.3% experienced one infection, and 21.7% had up to two infections. In the MMUD group, 17.9% of patients had no infections, 32.1% experienced one infection, and 27.4% experienced up to two infections. The haploidentical group showed that 14.3% of patients had no infections, 36.7% had one infection, and 28.6% experienced two infections.

### 3.5. Immune Reconstitution

The median time to immunosuppression discontinuation was similar among the three donor groups (*p* = 0.84): 185 days (IQR, 39–675) for HLA-matched, 197.5 days (IQR, 34–1883) for MMUD, and 186 days (IQR, 62–824) for haplo-HCT.

Immune reconstitution data were available for 185 (73%) patients (Figure 4). CD4+ T cells, CD8+ T cells, and IgG levels remained below normal before day +180 but gradually normalized beyond this point. CD4+ T cell recovery (>200 cells/μL) by day +180 was achieved in 51.6%, 36.7%, and 53.8% of patients in the HLA-matched, MMUD, and haploidentical groups, respectively. Similar trends were observed for CD8+ T cells and IgG recovery, with no significant differences between donor groups (*p* = 0.73).

## 4. Discussion

The present study provides a detailed examination of infectious complications and the dynamics of immune reconstitution in patients undergoing allo-HCT with PTCY-based prophylaxis stratified by donor type. The findings underscore the interplay between donor type, infectious risks, and immune recovery following allo-HCT with PTCY, adding valuable insights to the growing body of evidence on the use of this prophylaxis.

Our study reports low incidences of clinically relevant GVHD across all donor groups, aligning with rates reported in previous studies utilizing PTCY-based prophylaxis and supporting its use in clinical practice. In addition, despite the low incidence of cGVHD, disease control was comparable with previous publications [2,9,10]. Notably, the NRM trend was higher in MMUD (26.4%) and haploidentical (22.4%) recipients compared to HLA-matched transplants (11.8%, *p* = 0.05), supporting the selection of HLA-matched donors if available, when PTCY-based prophylaxis is used for GVHD prevention. Nonetheless, survival outcomes across donor groups were broadly comparable and aligned with previous related publications, as well as underscoring the feasibility of MMUD and haploidentical transplantation as a viable alternative in the PTCY era [21,22,23,24,25,26].

In contrast, our study describes higher infection rates and delayed immune recovery, both of which remain significant contributors to transplant morbidity in the context of PTCY-based approaches [27]. Data revealed that infection density was highest during the early post-transplant period, specifically within the first 100 days, with a median of one to two infections per patient and regardless of donor type. These results underscore the need for ongoing efforts to optimize infection prevention strategies, probably tailored to the immune reconstitution of patients, in this allo-HCT setting.

Consistent with data reported in previous studies [7,28,29], the incidence of BSI observed within the first 30 days post-allo-HCT was high. Notice that, while patients receiving grafts from HLA-matched donors exhibited a trend toward higher BSI incidence rates, the difference did not reach statistical significance. This finding apparently contrasts with previous reports suggesting that haploidentical donors may confer a higher infection risk due to the prolonged immune suppression required in this type of transplant [30,31,32]. Nevertheless, comparisons from these studies were mainly conducted with cohorts of patients who underwent allo-HCT from HLA-matched donors and other prophylaxis that did not contain PTCY. As reported in our analysis, BSI were caused by both gram-positive and gram-negative bacteria, probably due to the requirement of central line manipulation during PTCY administration, the incidence of gastrointestinal mucositis observed in this transplant setting, and the longer median time to neutrophil recovery induced by the cyclophosphamide administration [8].

Considering the high incidence of BSI, the optimization of central venous catheter care through standardized nursing protocols, together with the implementation of post-infusion G-CSF to reduce the duration of the aplastic phase, can be taken into consideration in allo-HCT performed with PTCY [33].

In addition, CMV reactivation was a notable viral complication, with the highest incidence observed in recipients of MMUD and haplo-HCTs. This aligns with the existing literature [34,35], which identifies MMUD and haploidentical transplants as a high-risk group due to increased HLA disparity and immunosuppressive intensity. These findings are consistent with previous reports [36] describing a high incidence of CMV reactivation in patients receiving PTCY-based prophylaxis. The median CMV reactivation time was 26 days in our cohort, comparable to other studies in allo-HCT recipients. Nevertheless, despite the high reactivation rates, the overall incidence of CMV disease remained low across all groups, reflecting the effectiveness of pre-emptive monitoring strategies and timely treatment.

During the study period, patients did not receive targeted anti-CMV prophylaxis with letermovir. However, letermovir prophylaxis was integrated into our practice for CMV-positive recipients undergoing transplantation with PTCY. Notably, preliminary results demonstrated that letermovir prophylaxis effectively decreased the incidence of CMV reactivation in patients receiving PTCY-based prophylaxis, and without increasing transplant toxicity [37,38]. Although letermovir prophylaxis is being extensively used in allo-HCT settings, the results in the present study are still of value as, unfortunately, its indication has not yet been universalized.

HHV-6 reactivation or infection was prevalent in our patients and even more frequent in haploidentical donor recipients, potentially indicating differences in immune recovery kinetics specific to this donor type. This reactivation, observed with a cumulative incidence of 22.4% in haploidentical transplants, aligns with previous studies reporting an increased risk for non-CMV herpesvirus infection, primarily driven by HHV-6 in haplo-HCT settings [39,40,41]. It is important to recognize that the incidence and morbidity of HHV-6 reactivation/infection in allo-HCT settings conducted with PTCY still needs further investigation. Limited studies have reported comparable incidences of this complication in haplo-HCTs. However, since HHV-6 reactivation and infection occurred also in allo-HCT performed from HLA-matched donors, exploring the morbidity induced by this infection in all allo-HCT with PTCY settings is needed.

Other viral infections, such as BK virus hemorrhagic cystitis, occurred in the context of PTCY-based prophylaxis. Results observed in our analysis align with previous data reported in a retrospective multicenter study conducted in adults undergoing haplo-HCT with PTCY and promoted by the Spanish Hematopoietic Cell Transplant and Cell Therapy Group (GETH-TC) [6], and a retrospective large study conducted by Princess Margaret Cancer Center that included patients undergoing allo-HCT from different donor types and PTCY [42]. The results, all together, support the need for enhancing prophylactic measures focused on hyperhydration, avoiding urinary retention during PTCY administration, and caution monitoring during the early post-transplant setting [43,44].

Respiratory tract infections represented another significant viral complication, with a steady increase in incidence during the first year post-allo-HCT, peaking 6 months after allo-HCT, though without significant differences between donor types. Rhinovirus was the most frequently identified pathogen in these cases (32.4%), followed by SARS-CoV-2 (26.8%). These findings align with global reports, emphasizing the need for preventive measures, including vaccination and early antiviral therapy [7,45].

Lastly, probable or proven fungal infections were observed in 12.7% of patients, predominantly pulmonary aspergillosis, which accounted for 62.8% of all events. These infections posed a significant challenge within the first 100 days post-allo-HCT, with no significant differences observed between donor types. The findings in haplo-HCT patients were consistent with data previously reported by the GETH-TC [6]. However, the data observed in MMUD and HLA-matched allo-HCT patients are particularly noteworthy, as most studies in PTCy-based settings to date have focused primarily on haplo-HCT cohorts.

At our institution, antifungal prophylaxis is routinely performed with fluconazole, except for transplant candidates with a prior history of probable or proven pulmonary aspergillosis or those requiring high doses of corticosteroids, in whom isavuconazole is administered. Given the results of this analysis, a dedicated investigation is currently underway to evaluate the effectiveness of our prophylactic approach against invasive fungal infections before considering potential modifications to our protocol.

Immune reconstitution metrics, including CD4+ and CD8+ T-cell recovery, revealed comparable patterns across the three donor groups. Notice additionally that the median time to immunosuppression discontinuation was also consistent, suggesting that donor type does not markedly influence the trajectory of immune recovery under standardized PTCY-based prophylaxis. The results correlate with previous publications, where immune reconstitution of PTCY patients was compared with that of patients who received tacrolimus/methotrexate or tacrolimus/sirolimus prophylaxis [46], and underscore the need for enhanced strategies to accelerate immune recovery, particularly in the context of preventing viral and fungal infections [27,46].

Although these results are considered of interest regarding the wide use of PTCY-based prophylaxis in allo-HCT settings, our study has limitations that should be recognized. The retrospective design and single-center setting may affect the generalizability of the findings. Additionally, the relatively small sample size may reduce the statistical power to detect subtle differences between groups, emphasizing the need for larger patient cohorts to validate these observations. The absence of standardized immune recovery assessments at predefined time points further restricts the ability to draw definitive conclusions regarding the kinetics of immune reconstitution. Future prospective studies with larger, multicenter cohorts are needed to validate these findings and explore the underlying mechanisms contributing to the observed differences in infectious complications.

## 5. Conclusions

Our study emphasizes the effectiveness of PTCY in reducing GVHD while maintaining similar immune reconstitution and survival outcomes across different donor types. However, the elevated incidence of BSIs, CMV, and respiratory viral infections—especially in MMUD and haploidentical transplants—indicates the necessity for improved infection prevention strategies, such as incorporating letermovir and broad-spectrum antimicrobial prophylaxis. Future research should focus on refining immune reconstitution monitoring and infection management to optimize transplant outcomes.

## Figures and Tables

**Figure 1 cancers-17-01109-f001:**
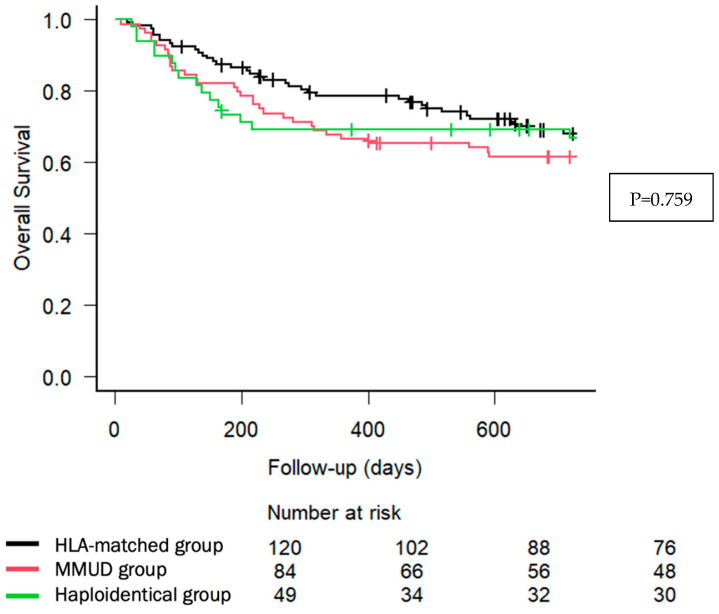
Two-year overall survival according to donor type. Kaplan–Meier curves describe OS in patients undergoing allo-HCT with PTCY-based prophylaxis, stratified by donor type: HLA-matched (black line), MMUD (red line), and haploidentical donors (green line). The *x*-axis represents follow-up time in days, while the *y*-axis indicates overall survival probability.

**Figure 2 cancers-17-01109-f002:**
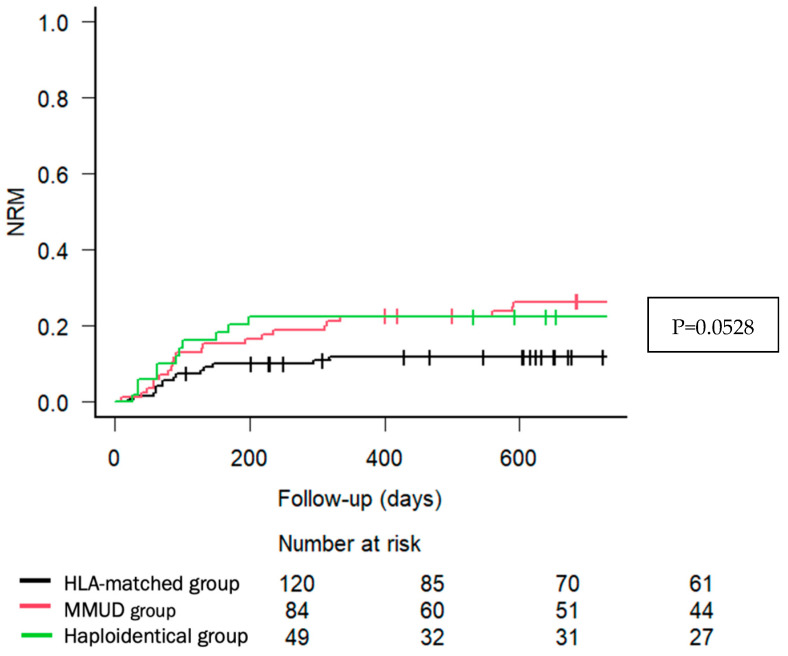
Two-year non-relapse mortality according to donor type. Cumulative incidence of non-relapse mortality (NRM) by donor type. The Kaplan–Meier curves represent NRM over time in patients undergoing allo-HCT with post-transplant PTCY-based prophylaxis, stratified by donor type: HLA-matched (black line), MMUD (red line), and haploidentical donors (green line). The *x*-axis indicates follow-up time in days, while the *y*-axis represents the cumulative incidence of NRM.

**Figure 3 cancers-17-01109-f003:**
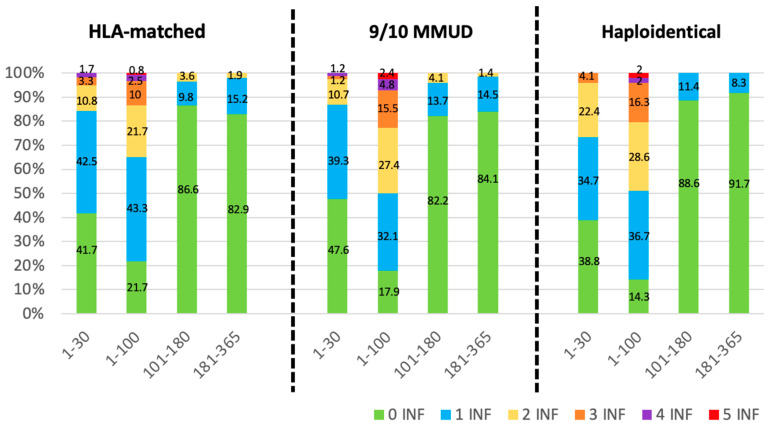
Infection density analyses. Proportion of patients experiencing infections at different time points after allo-HCT (1–30, 1–100, 101–180, and 181–365 days) stratified by donor type (HLA-matched, 9/10 MMUD, and haploidentical). The color scale indicates infection count per patient: 0 INF (green): no infections; 1 INF (blue): one infection; 2 INF (yellow): two infections; 3 INF (orange): three infections; 4 INF (purple): four infections; 5 INF (red): five infections.

**Figure 4 cancers-17-01109-f004:**
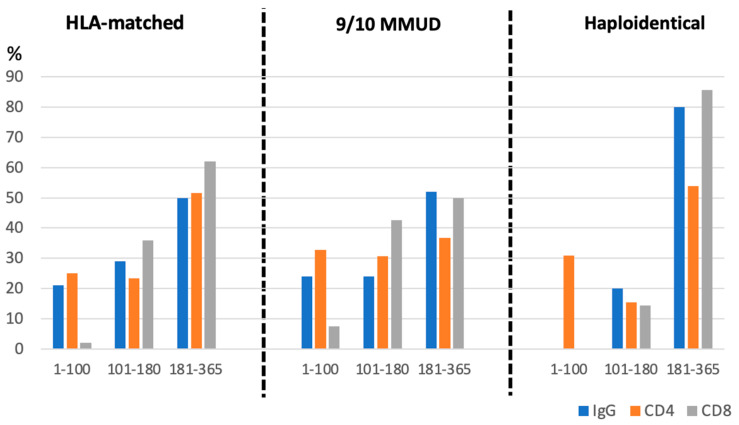
Immune reconstitution analyses.

**Table 1 cancers-17-01109-t001:** Baseline patient characteristics and allo-HCT information.

	HLA-Matched (*n* = 120)	9/10 MMUD (*n* = 84)	Haploidentical (*n* = 49)	*p* Value
**Age Median (range)** **≥65 years**	53 (23–70) 22 (18.3)	52 (18–68) 12 (14.3)	53 (20–70) 10 (20.4)	0.464 0.325
**Sex** Female	49 (40.8)	33 (39.3)	23 (46.9)	0.674
**Baseline Diagnosis** AML MDS MPN ALL Lymphoproliferative disorders CML PCD Others	38 (31.7) 23 (19.2) 9 (7.5) 21 (17.5) 13 (10.8) 1 (0.8) 13 (10.8) 2 (1.7)	28 (33.3) 20 (23.8) 4 (4.8) 17 (20.2) 9 (10.7) 4 (4.8) 2 (2.4) 0	24 (49.0) 3 (6.1) 0 8 (16.3) 13 (26.5) 0 1 (2.0) 0	N/A
**HCT-CI** > 3	38 (31.7)	15 (17.9)	8 (16.3)	0.028
**Karnofsky Performance Status** 70–80% ≥90%	28 (23.4) 92 (76.6)	23 (27.4) 61 (72.6)	10 (20.4) 39 (79.6)	0.675
**CMV Risk Status** Low risk Intermediate High	17 (14.16) 91 (75.83) 12 (10.01)	8 (9.52) 73 (86.91) 3 (3.57)	4 (8.16) 40 (81.63) 5 (10.2)	0.48
**Donor/Recipient characteristics** Donor female -> male patient	3 (2.5)	1 (1.2)	0	0.467
**Intensity** Myeloablative Reduced intensity	46 (38.3) 74 (61.7)	37 (44) 47 (56)	20 (40.8) 29 (59.2)	
**Conditioning Regimen (Extended)** Flu/Bu (4) (± TBI) Flu/TBI (12 Gy) TBF Cy/Flu/TBI (2 Gy) Flu/TBI (8 Gy) Flu/Bu (3) Flu/Mel Sequential RIC allo-HCT Other	23 (19.1) 18 (15) 4 (3.3) 1 (0.8) 8 (6.7) 45 (37.5) 13 (10.8) 6 (5.0) 3 (2.4)	21 (25.0) 14 (16.7) 1 (1.2) 1 (1.2) 3 (3.6) 30 (35.7) 2 (2.4) 6 (7.1) 6 (7.1)	11 (22.4) 6 (12.2) 1 (2.0) 11 (22.4) 1 (2.0) 17 (34.7) 0 1 (2.0) 1 (2.0)	N/A
**Median Follow-Up: Months (IQR)**	25 (10.16–51.63)	33.06 (7.66–60.96)	30.26 (5.52–68)	

MMUD, mismatched unrelated donors; AML, acute myeloid leukemia; MDS, myelodysplastic syndrome; MPN, myeloproliferative neoplasm; ALL, acute lymphoblastic leukemia; CML, chronic myeloid leukemia; PCD, plasma cell dyscrasia; N/A, not applicable; HCT-CI, Hematopoietic Cell Transplantation Comorbidity Index; CMV, citomegalovirus; Flu, fludarabine; Bu, busulfan; TBI, total body irradiation; TBF, thiotepa/busulfan/fludarabine; Cy, cyclophosphamide; Mel, mefalan; RIC, reduced-intensity conditioning.

**Table 2 cancers-17-01109-t002:** Main post-transplant outcomes.

	HLA-Matched (*n* = 120)	9/10 MMUD (*n* = 84)	Haploidentical (*n* = 49)	*p* Value
**Median Days of Transplant Hospitalization (IQR)** **Post-Transplant Information:** SOS TMA	30 (17–79) 1 (0.8) 4 (3.3)	30 (9–155) 2 (2.4) 6 (7.1)	30 (17–169) 1 (2) 4 (8.2)	0.62 0.467 0.674
**Engraftment Information** Median days neutrophil engraftment (IQR) Median days platelet engraftment (IQR) Primary graft failure (%)	19 (11–62) 18 (6–128) 5 (4.1)	19 (12–89) 18 (8–141) 6 (7.1)	19 (14–103) 18 (9–63) 1 (0.02)	0.692 0.651 0.467
**Cumulative Incidence Infection Complications:** Bacterial Bloodstream Infections: - Day + 30 - Day + 100 - Day + 180 - Day + 365 CMV Reactivation: - Day + 30 - Day + 100 - Day + 180 - Day + 365 CMV Disease: - Day + 30 - Day + 100 - Day + 180 - Day + 365 Grade 2–4 BK Hemorrhagic Cystitis: - Day + 30 - Day + 100 - Day + 180 - Day + 365 VHH6 Infection: - Day + 30 - Day + 100 - Day + 180 - Day + 365 Respiratory Viral Infection: - Day + 30 - Day + 100 - Day + 180 - Day + 365 Fungal Infection: - Day + 30 - Day + 100 - Day + 180 - Day + 365	49.2 (39.9–57.8) 53.3 (44.0–61.8) 54.2 (44.8–62.6) 56.8 (47.4–65.2) 9.2 (4.8–15.2) 39.2 (30.4–47.8) 40.9 (32.0–49.5) 40.9 (32.0–49.5) 1.7 (0.3–5.4) 5.8 (2.6–11.0) 6.7 (3.1–12.1) 6.7 (3.1–12.1) 5.0 (2.0–10.0) 7.6 (3.7–13.2) 10.3 (5.6–16.6) 10.3 (5.6–16.6) 2.5 (0.7–6.6) 7.5 (3.7–13.1) 9.2 (4.9–15.2) 11.9 (6.8–18.4) 7.5 (3.7–13.1) 10.8 (6.1–17.2) 19.2 (12.7–26.7) 26.9 (19.2–35.1) 5.9 (2.6–11.1) 6.7 (3.1–12.2) 6.7 (3.1–12.2) 11.1 (6.2–17.6)	38.1 (27.7–48.4) 46.4 (35.4–56.7) 50.0 (38.8–60.2) 51.2 (40.0–61.3) 8.3 (3.6–15.5) 59.5 (48.1–69.2) 61.9 (50.5–71.4) 61.9 (50.5–71.4) 2.4 (0.5–7.5) 6.0 (2.2–12.4) 8.3 (3.6–15.5) 11.9 (6.1–19.9) 2.4 (0.5–7.6) 22.3 (13.9–32.0) 23.7(15.0–33.6) 23.7(15.0–33.6) 8.3 (3.6–15.5) 9.5 (4.4–17.0) 10.7 (5.2–18.5) 11.9 (6.1–19.9) 4.8 (1.5–10.9) 9.5 (4.4–17.0) 17.9 (10.5–26.8) 25.0 (16.3–34.7) 5.2 (1.7–11.8) 13.0 (6.6–21.6) 13.0 (6.6–21.6) 14.3 (7.6–23.1)	34.7 (21.7–48.0) 38.8 (25.1–52.2) 38.8 (25.1–52.2) 38.8 (25.1–52.2) 20.4 (10.4–32.7) 55.1 (40.0–67.9) 55.1 (40.0–67.9) 55.1 (40.0–67.9) 4.1 (0.7–12.5) 6.1 (1.6–15.3) 8.2 (2.6–18.0) 8.2 (2.6–18.0) 4.1 (0.7–12.6) 14.8 (6.4–26.5) 14.8 (6.4–26.5) 14.8 (6.4–26.5) 14.3 (6.2–25.6) 22.4 (11.9–35.0) 24.5 (13.5–37.2) 24.5 (13.5–37.2) 8.2 (2.6–18.0) 16.3 (7.6–28.0) 20.4 (10.4–32.7) 24.6 (13.5–37.5) 6.4 (1.6–15.9) 8.5 (2.7–18.7) 10.6 (3.8–21.4) 12.8 (5.1–24.2)	0.073 0.033 0.449 0.056 0.068 0.744 0.640
**Cumulative Incidence GVHD [% (95% CI)]** Grade 2–4 acute GVHD at day +100 Grade 3–4 acute GVHD at day +100 Moderate/severe chronic GVHD at 2 years	19.2 (12.7–26.7) 6.7 (3.1–12.1) 3.6 (1.2–8.4)	31.0 (21.4–41.0) 8.3 (3.6–15.5) 13.6 (6.9–22.6)	22.4 (11.9–35.0) 4.1 (0.7–12.5) 9.6 (3.0–21.0)	0.055 0.848 0.062
**Median Days to IS Discontinuation (IQR)**	185 (39–675)	197.5 (34–1883)	186 (62–824)	0.842
**Main Post-Transplant Outcomes (% (95% CI))** Overall survival 2 years Relapse-free survival 2 years Non-relapse mortality 2 years Cumulative incidence of relapse 2 years GRFS 2 years	68.1 (58.5–75.9) 55.2 (45.6–63.8) 11.8 (6.8–18.4) 33.0 (24.5–41.7) 51.0 (41.5–59.7)	61.6 (50.3–71.1) 53.1 (41.9–63.2) 26.4 (17.4–36.3) 20.4 (12.5–29.7) 42.3 (31.5–52.6)	66.8 (51.6–78.2) 54.4 (39.3–67.2) 22.4 (11.9–35.0) 23.2 (12.3–36.1) 46.0 (31.5–59.4)	0.759 0.995 0.0528 0.0972 0.55

## Data Availability

Data sharing would be only considered after specific request.

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
