# Peer review of "Post-Transplant Cyclophosphamide-Based Prophylaxis and Its Impact on Infectious Complications and Immune Reconstitution According to Donor Type"

_cancers, 2025, doi:10.3390/cancers17071109_

Round 1
Reviewer 1 Report
Comments and Suggestions for Authors
The authors evaluated infectious complications and immune reconstitution in 253 adults post-
allogeneic peripheral blood allogeneic hematopoietic cell transplantation (allo-HCT) with post-
transplant cyclophosphamide (PTCY) - based GVHD prophylaxis. Grafts were from 20
HLA-matched donors (47.4%), mismatched unrelated donors (MMUD, 33.2%), or haploidentical 21 donors (19.4%). The median time to neutrophil engraftment was 19 days. The cumulative incidence of acute and chronic GVHD, immunosuppression duration and post-transplant out-
comes, are reported to be similar across donor types. They made a comprehensive study of post-HSCT infectious complications, as well as acute and chronic GvHD and report that at day +30 Cum.Inc of bacterial bloodstream infections (BSI) was highest in HLA-matched transplants (49.2%, p=0.073). Cytomegalovirus (CMV) reactivation occurred between days +30 and +100, with the highest Cum.Inc in MMUD (59.5%, p=0.033). HHV-6 reactivation was more frequent in haploidentical transplants (22.4%, p=0.068). BK virus-associated hemorrhagic cystitis showed a trend toward higher incidence in MMUD (22.3%, p=0.056). Respiratory and fungal infections were most common during the first 100 days, without major differences according to donor type. By day +180, immune reconstitution was achieved in most patients, with 30
normalization of CD4+ T cells, CD8+ T cells, and IgG levels, independent of donor type. Patients after allo-HCT with PTCY-based prophylaxis experience a high infectious complications early
post-transplant up to 6 months as immune reconstitution progressesindependent of donor type used.
This is an excellent summary of complications post-allo HSCT with PTCY as GvHD prophylaxis. The study highlights the possibilities of infectious complications, which are increased compared to other GvHD-prophylaxis regimens.
I have actually only one question:
Did you see differences in patients with Letermovir as CMV-prophylaxis compared to the earlier pre-emptive therapy protocols? Accure CMV-reactivation in lower frequency and at later time points also in the context of PTCY prophylaxis?
Author Response
The authors evaluated infectious complications and immune reconstitution in 253 adults post- allogeneic peripheral blood allogeneic hematopoietic cell transplantation (allo-HCT) with post- transplant cyclophosphamide (PTCY) - based GVHD prophylaxis. Grafts were from 20 HLA-matched donors (47.4%), mismatched unrelated donors (MMUD, 33.2%), or haploidentical 21 donors (19.4%). The median time to neutrophil engraftment was 19 days. The cumulative incidence of acute and chronic GVHD, immunosuppression duration and post-transplant out- comes, are reported to be similar across donor types. They made a comprehensive study of post-HSCT infectious complications, as well as acute and chronic GvHD and report that at day +30 Cum.Inc of bacterial bloodstream infections (BSI) was highest in HLA-matched transplants (49.2%, p=0.073). Cytomegalovirus (CMV) reactivation occurred between days +30 and +100, with the highest Cum.Inc in MMUD (59.5%, p=0.033). HHV-6 reactivation was more frequent in haploidentical transplants (22.4%, p=0.068). BK virus-associated hemorrhagic cystitis showed a trend toward higher incidence in MMUD (22.3%, p=0.056). Respiratory and fungal infections were most common during the first 100 days, without major differences according to donor type. By day +180, immune reconstitution was achieved in most patients, with 30 normalization of CD4+ T cells, CD8+ T cells, and IgG levels, independent of donor type. Patients after allo-HCT with PTCY-based prophylaxis experience a high infectious complications early
post-transplant up to 6 months as immune reconstitution progressesindependent of donor type used.
This is an excellent summary of complications post-allo HSCT with PTCY as GvHD prophylaxis. The study highlights the possibilities of infectious complications, which are increased compared to other GvHD-prophylaxis regimens.
I have actually only one question:
Did you see differences in patients with Letermovir as CMV-prophylaxis compared to the earlier pre-emptive therapy protocols?
Thank you for your question. Letermovir prophylaxis has been incorporated into our clinical practice for all CMV-positive recipients undergoing transplantation with PTCY in 2022. For the study conduction, no patients receiving letermovir prophylaxis were included because the sample size was very low.
Our experience, as well as findings from previous studies, confirm than letermovir prophylaxis significantly reduces the incidence of CMV reactivation in patients receiving PTCY-based prophylaxis, without increasing transplant-related toxicity. However, this specific aspect cannot be directly investigated in this study due to the lack of patients receiving letermovir prophylaxis. A detailed analysis of this effect has been reported in a recent study conducted by our group. (Brusosa M, Ruiz S, Monge I, Solano MT, Rosiñol L, Esteve J, Carreras E, Marcos MÁ, Riu G, Carcelero E, et al. Impact of letermovir prophylaxis in CMV reactivation and disease after allogeneic hematopoietic cell transplantation: a real-world, observational study. Ann Hematol. 2024;103(2).
Considering your comment we have highlighted the following information included in the Methods and Results Sections:
“No patient received letermovir for cytomegalovirus (CMV) prophylaxis.”
“During the study period, patients did not receive targeted anti-CMV prophylaxis with letermovir. However, letermovir prophylaxis was integrated into our practice for CMV-positive recipients undergoing transplantation with PTCY. Notably, preliminary results demonstrated that letermovir prophylaxis effectively decreased the incidence of CMV reactivation in patients receiving PTCY-based prophylaxis, and without increasing transplant toxicity (36,37). Although letermovir prophylaxis is being extensively used in allo-HCT settings, the results in the present study are still of value as unfortunately, its indication has not yet been universalized.”
Accure CMV-reactivation in lower frequency and at later time points also in the context of PTCY prophylaxis?
Thank you for your question. Results observed in a large retrospective study conducted by the CIBMTR and published in Blood (Teira P, Battiwalla M, Ramanathan M, Barrett AJ, Ahn KW, Chen M, et al. Early cytomegalovirus reactivation remains associated with increased transplant-related mortality in the current era: A CIBMTR analysis. Blood. 2016;127(20)) demonstrated that patients receiving PTCY had increased incidences of CMV reactivation than patients undergoing allo-HCT with other prophylaxis that did not contain PTCY.
However, our study cannot address this question as only includes patients transplanted with PTCY-based prophylaxis. Nevertheless, the median of days to CMV reactivation in PTCY-based patients was 26 days (similar to what has been reported in other cohorts of allo-HCT patients who received different GVHD prophylactic strategies).
Unfortunately, the present analysis focuses its effort on exploring the incidence of the first infectious event. No second (or more) reactivations or second infections have been recorded during the study collection for its analysis. Nevertheless, we can speculate that later reactivations might be less frequent in PTCY-based patients due to the low incidence of acute and chronic GVHD documented in these patients. Hence, an earlier discontinuation of immunosuppression and the low incidence of corticosteroid treatment requirements might positively influence on CMV reactivations in PTCY-based patients.
Considering your comment we have highlighted the following information included in the Discussion Section:
“These findings are consistent with previous reports (36) describing a high incidence of CMV reactivation in patients receiving PTCY-based prophylaxis. The median CMV reactivation time was 26 days in our cohort, comparable to other studies in allo-HCT recipients.”
Reviewer 2 Report
Comments and Suggestions for Authors
This is a rather well designed and performed study.There are no really significant findings,which may be due to that there are not enough patients.To say there are no differences between the three groupos,you need even more patients.This should be acknowledeged in the paragraph in the discussion referring to the shortcomings of this article.
In abstract add that differences" tended to be"BSI and HHV6" that were ns ie p>0.05.NRM was different,which should be added in the abstract.
Table 1, explain the abbreviations.
Define neutrophil and platlet engraftment.
Regarding Karnofsky it look odd with 70-80%.I suppose all others had >=90%.Clarifye!
Among HLA identical donor recipient pairs.How many were HLA identical siblings and how many were MUD?Among the MUD what was HLA compatibility 12/12 10/10 ?
Table 2 need to be corrected. Respiratory viral infections and Fungal infection figures are overlapping,which is sloppy.
Fig.2 Add p-value in the figure.
Line 178 ns write "tended to be!"
In fig 3 and fig 4 change accordingly.Compare HLA matched vs 9/10 MUD vs Haplo 1-30 days,
1-100 days vs 101-180 days and 181-365 days.For fig 4 thre groups 1-100,101-180 and 181-365
Discussion Line 252 elevated compared to what expand!
Author Response
This is a rather well designed and performed study. There are no really significant findings, which may be due to that there are not enough patients. To say there are no differences between the three groupos, you need even more patients. This should be acknowledeged in the paragraph in the discussion referring to the shortcomings of this article.
We sincerely appreciate your comments. You are absolutely correct in pointing out that the sample size may limit the ability to detect differences between the three groups. In response to your suggestion, we have revised the limitations section of the discussion to explicitly acknowledge this issue. We now highlight that the relatively small cohort may reduce statistical power and emphasize the need for larger, multicenter studies to validate our findings.
“Although results are considered of interest regarding the widely use of PTCY-based prophylaxis in allo-HCT settings, our study has limitations that should be recognized. The retrospective design and single-center setting may affect the generalizability of the findings. Additionally, the relatively small sample size may reduce the statistical power to detect subtle differences between groups, emphasizing the need for larger patient cohorts to emphasizing the need for larger patient cohorts to validate these observations. Future prospective studies with larger, multicenter cohorts are needed to validate these findings and explore the underlying mechanisms contributing the observed differences in infectious complications.”
In abstract add that differences" tended to be"BSI and HHV6" that were ns ie p>0.05. NRM was different, which should be added in the abstract.
We have now clarified that differences in BSI and HHV6 "tended to be" non-significant (p > 0.05) and explicitly added the difference in NRM to the abstract.
“The day +30 Cum.Inc of bacterial bloodstream infections (BSI) tended to be higher in HLA-matched transplants (49.2%, p=0.073), while HHV-6 reactivation showed a trend toward higher frequency in haploidentical transplants (22.4%, p = 0.068).”
Table 1, explain the abbreviations.
We have included explanations for all abbreviations to ensure clarity.
Define neutrophil and platlet engraftment
While these definitions were originally in the Supplementary Materials, we have now incorporated them into the main text for better accessibility.
“Engraftment after allo-HCT was defined as the presence of an absolute neutrophil count greater than ≥0.5 x 109/L on the first of three consecutive days. Platelet recovery was defined as a sustained platelet count > 20 x 109/L (1st of 3 days) without platelet transfusion for 7 days.”
Regarding Karnofsky it look odd with 70-80%.I suppose all others had >=90%.Clarifye!
We acknowledge that the distribution may have appeared unclear. We have now clarified that all other patients had a Karnofsky score of ≥90% in the table.
Among HLA identical donor recipient pairs. How many were HLA identical siblings and how many were MUD? Among the MUD what was HLA compatibility 12/12 10/10 ?
Thank you for your insightful question. We have now clarified the information regarding HLA compatibility among the donor-recipient pairs. Among the HLA-matched donor-recipient pairs, 28 patients (23.3%) received grafts from an HLA-matched sibling donor, while 92 patients (72.7%) underwent transplantation with a 10/10 HLA-matched unrelated donor (MUD).
Regarding your question about HLA compatibility among the MUD group, DPB1 match / mismatched was not accounted during the study conduction, and all MUD had a 10/10 HLA match.
Considering your comment, the following information has been included into the Results Section:
“Among the 10/10 HLA-matched donor-recipient pairs, 28 patients (23.3%) received grafts from 10/10 HLA-matched sibling donor, while 92 patients (72.7%) underwent transplantation with 10/10 HLA-matched unrelated donor (MUD).”
Table 2 need to be corrected. Respiratory viral infections and Fungal infection figures are overlapping, which is sloppy.
Thank you for your comment. The figures for respiratory viral infections and fungal infections have been corrected to eliminate any overlap.
Fig.2 Add p-value in the figure.
We have now included the p-value in the figure.
Line 178 ns write "tended to be!"
We have revised "ns" to "tended to be" as suggested.
In fig 3 and fig 4 change accordingly. Compare HLA matched vs 9/10 MUD vs Haplo 1-
30 days, 1-100 days vs 101-180 days and 181-365 days. For fig 4 thre groups 1-100,101-180 and 181-365.
Thank you for your suggestion. We understand the importance of maintaining consistency in time intervals for comparison. However, in Figure 4, which represents immune reconstitution, we did not include the 1–30 day interval because immune recovery is not observed before day +60 in any case. For this reason, we believe that adding this time frame would not provide meaningful information.
Nevertheless, to facilitate comparison, we have included the 1–100 day interval in both Figures 3 and 4 as suggested. This ensures alignment between the figures while maintaining a biologically relevant representation of immune reconstitution.
Discussion Line 252 elevated compared to what expand!
Thank you for your suggestion. We appreciate your suggestion and have clarified the discussion line 252.
“In contrast, our study describes higher infection rates and delayed immune recovery, both of which remain significant contributors to transplant morbidity in the context of PTCY-based approaches(27).”